# Poor Spontaneous Recovery of Aphemia Accompanied by Damage to the Anterior Segment of the Left Arcuate Fasciculus: A Case Report

**DOI:** 10.3390/brainsci12091253

**Published:** 2022-09-16

**Authors:** Qiwei Yu, Wenjun Qian

**Affiliations:** Department of Rehabilitation Medicine, The Affiliated Suzhou Hospital of Nanjing Medical University, Suzhou 215008, China

**Keywords:** aphemia, stroke, arcuate fasciculus, diffusion tensor imaging

## Abstract

Aphemia is a rare and special type of speech disorder, and the mechanisms underlying the occurrence and recovery remain unclear. Here, we present a clinical case of poor spontaneous recovery of aphemia, with the anterior segment of the left arcuate fasciculus server damaged and the posterior segment intact, as detected by diffusion tensor imaging. Aphemia could be caused by the disruption of the cortical and subcortical language circuits. In particular, our data support the view that damage to the anterior segment of the left arcuate fasciculus may result in poor spontaneous recovery from speech production deficits and that an intact posterior segment seems to be crucial for supporting residual language comprehension ability in patients with post-stroke aphasia. Collectively, these data imply the importance of the left arcuate fasciculus during recovery from the language disorder in the subacute stage of stroke.

## 1. Introduction

Aphemia is also known as simple aphasia, ataxic aphasia, cortical anarthria, or pure word mutism and represents a rare and special type of articulatory disorder [1]. Patients with aphemia are initially mute, with the subsequent incomplete recovery of speech ability characterized by impaired articulation and cadence with intact vocabulary, syntax, and grammar [2]. This condition was first reported by Broca in 1861 but is rarely encountered in clinical practice. Broca initially used the term “Aphemia” to describe a singular symptom of language deficits in a patient who presented with the loss of speech but without the impairment of language [3]. They retained the ability to move the faciolingual territories employed in speech. Hence, aphemia was regarded as a disturbance in the organization of articulatory and motor aspects of speech, with intact grammar, reading, and writing [1]. Subsequently, Trousseau coined the term “aphasia” and claimed that “aphasia” was more appropriate to describe language disorders extended beyond articulation to include comprehension, reading, and writing [1,4]. Therefore, “Aphemia” was reserved for isolated speech deficits [2].

The mechanisms of aphemia are poorly understood. Several previous reports [5,6,7] have speculated that damage to the cortical areas may be an important underlying factor for aphemia. As well, it was also postulated that injury to the subcortical fiber tracts could also cause aphemia [3,5]. However, this topic has not been reported on, and little is known about the role of white matter pathways in aphemia. Herein, we describe a patient who presented with poor spontaneous recovery of aphemia secondary to a left hemisphere infarction that occurred two months previously. In particular, we aimed to explore the occurrence and recovery mechanisms underlying this special type of language deficit from the perspective of white matter pathways.

## 2. Case Description

A 64-year-old right-handed male patient, with no history of brain damage, was referred to the emergency department because of a 3 h history of right limb weakness and speech disorder. Although conscious, his spontaneous speech and auditory comprehension were both noticeably impaired. He was unable to produce sound. The manual muscle testing (MMT, modified Lovett muscle test) [8,9] scores were 0/5 for the right upper limb, 4/5 for the right lower limb, and 5/5 for the left leg. His National Institutes of Health Stroke Scale (NIHSS) [10] score was 15, including 4 for upper limbs, 1 for lower limbs, 2 for language, and 2 for articulation. An immediate computed tomography (CT) of the brain and angiography (CTA) detected an ischemic lesion in the left hemisphere, with severe stenosis or occlusion of the left internal carotid artery and the left middle cerebral artery. He was diagnosed with acute ischemic stroke and received medication treatments including recombinant tissue plasminogen activator (rt-PA), tirofiban, aspirin, and clopidogrel. He also received surgical treatment, including left internal carotid artery balloon dilation and stenting and left middle cerebral artery stent thrombectomy in the neurology department. Subsequent magnetic resonance imaging (MRI) of the brain, completed two days later, confirmed the cerebral infarction in the left hemisphere with a left basal ganglia hemorrhage (Figure 1A). He survived the acute period of hospitalization and was transferred to a rehabilitation facility. After two months of consecutive physical therapy and drug treatments (but not speech-language therapy), the motor function of his right limbs significantly improved, and his auditory comprehension returned to almost normal. He was able to write with his left hand and perform calculations. His ability to swallow, smile, and blow air remained despite an obvious right central facial palsy. The soft palate did not deviate, and the tongue moved flexibly. However, he was unable to produce sound or even cough. There was no evidence of vocal cord paralysis, and his wife denied any relevant previous history of recurrent laryngeal nerve or vocal cord lesion.

Next, we used the Chinese version of the western aphasia battery (WAB) [11,12] to assess his language ability. We found that the patient had attained certain improvements in his language functions (spontaneous speech: 0 percentile, speech comprehension: 77.5 percentile, repetition: 0 percentile, naming: 0 percentile, reading: 0 percentile, and writing: 88.5 percentile). The aphasia quotient (AQ) was 31.0. According to the report of Broca and the description of Oliveira-Souza Rd et al. [3], he was diagnosed with aphemia. A follow-up brain MRI, carried out two months after the onset of the stroke, demonstrated an old ischemic lesion in the left hemisphere (Figure 1B).

## 3. Diffusion Tensor Imaging and Tractography Evaluation

All MRI images were acquired by a Siemens Skyra 3.0T MR scanner equipped with a standard 12-channel phased-array head coil. For diffusion tensor imaging (DTI) sequences, a single-shot echo planar imaging sequence was acquired in 47 contiguous slices parallel to the anterior-posterior plane for each of 64 non-collinear diffusion sensitizing gradients. The parameters were as follows: time of repetition = 4300 ms, time of echo = 92.00 ms, flip angle = 90, acquisition matrix = 96 × 96 mm^2^, reconstructed to matrix = 128 × 128 matrix, field of view = 768 × 768 mm^2^, b = 0, 1000 s/mm^2^, number of excitations = 2, slice thickness/slice spacing = 3/0 mm.

We used the FMRIB Software Library (FSL) [13] 5.0.9 software package (http://fsl.fmrib.ox.ac.uk/fsl/fslwiki/FSL accessed on 20 June 2022) for preprocessing and the analysis of diffusion data. The MRI datasets, including the three-dimensional T1-weighted image series and DTI series, were spatially normalized in the MNI152 atlas space (the MNI152_T1_2mm brain template). The Diffusion Toolkit 0.6.4.1 [14] was used for diffusion imaging data reconstruction and fiber tracking. We also used TrackVis (http://www.trackvis.org/, Charlestown, MA, United States; accessed on 20 June 2022) version 0.6.1 software to manually draw regions of interest (ROI) and perform fiber track visualization. A three-ROI approach was used to accomplish virtual dissection of the bilateral arcuate fasciculus (AF) based on the deterministic fiber-tracking approach: a Frontal ROI (ROI 1, the green 2D disk) was manually placed on the coronal slice at the entrance to the frontal lobe (anterior to the central sulcus), a Temporal ROI (ROI 3, the blue 2D disk) was manually placed on the axial slice at the entrance to the temporal lobe (below the Sylvian fissure), and a Parietal ROI (ROI 2, the red 3D sphere) was manually placed tangent to the inferior parietal cortex [15,16]. (Figure 2). The anterior segment of the AF was defined when the tract passed through both the ROI 1 and ROI 2, but not the ROI 3, the posterior segment was defined when the tract passed through both the ROI 2 and ROI 3, but not the ROI 1, and the long segment was defined when the tract passed through both the ROI 1 and ROI 3, but not the ROI 2. All ROIs for both hemispheres were obtained on fractional anisotropy images. Fiber tracking was initiated with fractional anisotropy (FA) > 0.20 and an angle threshold of 35°.

As shown in Figure 2 and Figure 3, the anterior segment of the left AF could not be reconstructed by using the DTI tractography due to severe injury (i.e., a disconnection between Broca’s and Geschwind’s areas). The anterior portion of the long segment was damaged, while the posterior segment was intact (i.e., a connection between Geschwind’s and Wernicke’s areas). In addition, the infarct lesion in the left hemisphere involved the opercular part (the red block) and the triangular part of the left inferior frontal gyrus (the green block).

## 4. Discussion

In this report, we described a patient who presented with poor spontaneous recovery of aphemia secondary to an ischemic stroke in the left hemisphere two months previously. Diffusion tensor imaging and tractography revealed that the anterior segment of the left arcuate fasciculus (AF) was not reconstructed due to severe injury, and the anterior portion of the long segment was damaged, while the posterior segment was intact. Exploring the brain mechanisms of aphemia from the perspective of white matter pathways is a unique approach and of significant interest. In addition to the consequence of the damage incurred by the left frontal lobe, we speculated that injury to the anterior segment of the left AF might be one of the most important factors underlying the poor spontaneous recovery of the patient’s speech production ability and that the intact posterior segment might support his relatively reserved language comprehension performance.

It has been reported that patients with aphemia commonly present with severe difficulties in terms of expressive language, including spontaneous speech, naming, repetition, and reading, while auditory comprehension and writing are relatively reserved [17]. The occurrence and recovery mechanisms of aphemia remain highly unclear. It was speculated that this condition could be mainly caused by lesions affecting the cortical areas such as the lower part of the left primary motor cortex and the contiguous premotor cortex [5] or the posterior third of the left inferior frontal gyrus [6,7]. Especially, it was reported that the lesion responsible for aphemia was usually located in the opercular portion of the inferior frontal gyrus [3,5]. The frontal operculum consists of the kinetic formula which supports the automatic conversion of verbal language into speech. Therefore, damage to this special cortex area will be expected to cause speech impairment without compromising language, faciolingual motility, and the ability to produce vocal sounds.

With the development of diffusion imaging techniques over the last few decades, it has become established that the white matter pathway may play an important role in language deficits and outcome prediction after stroke [18,19,20]. Due to its special anatomical location, the AF is considered the main neural pathway connecting Broca’s area and Wernicke’s area, and it has been particularly associated with language-related white matter fiber bundles. In 2005, Catani et al. [21] reported the two parallel pathways model of the AF, including a direct pathway that encompasses the classical connection between Broca’s and Wernicke’s area and an indirect pathway that consists of an anterior segment and a posterior segment. Specifically, the anterior segment links Broca’s area with Geschwind’s area (i.e., the inferior parietal lobule), while the posterior segment connects Geschwind’s area with Wernicke’s area. (Figure 4). Functionally, it was suggested that the direct pathway is involved in phonological processing (such as automatic repetition), and the indirect pathway is related to lexical-semantic processing (such as auditory comprehension and vocalization of semantic content). Therefore, they speculated that injury to the anterior segment of the AF would disconnect the link between the Broca’s area and the parietal region and result in a failure to vocalize semantic content, while injury to the posterior segment would disconnect the link between the Wernicke’s area and the parietal region and lead to a failure of auditory semantic comprehension [21]. Accordingly, the AF may support certain language skills, including speech production, semantic comprehension, naming accuracy, repetition, and reading [22,23].

Accumulating studies have indicated a strong relation between damage to the AF and various language deficits. Song et al. [24] reported that a lesion involving Broca’s area and the anterior segment of the AF would lead to Broca-like conduction aphasia, whereas a lesion involving Wernicke’s area and the posterior segment of the AF would lead to Wernicke-like conduction aphasia. Geva et al. [25] reported that injury to the left AF was correlated with impairments in word repetition, sentence comprehension, object naming, and homophone and rhyme judgment. Hope et al. [26] found that damage to the AF was significantly associated with deficits in speech fluency and naming accuracy. However, in a study of chronic stroke patients, Fridriksson et al. [27] argued that the components of language comprehension and production cannot be easily separated, as speech fluency involved factors related to both comprehension and production. Indeed, given the distribution of vascular lesions, a stroke lesion will destroy both proximal grey and white matter, which results in the common coexistence of damage to cortical language areas and injury to the AF. Hence, it is difficult to illustrate the accurate role of AF in language processing.

As mentioned, aphemia is a special type of speech disorder and is mainly characterized by expressive language disorder. Exploring the neural mechanisms underlying the language impairments in this special population helps to reveal the role of the AF in language processing. To the best of our knowledge, this is the first case report to explore the brain mechanisms underlying aphemia from the perspective of white matter pathways. Based on anatomy—specifically, that the anterior segment of the AF extends from Broca’s territory [28,29]—it follows that damage to this tract would negatively affect speech production, at least in theory. In a study including 64 patients with chronic non-fluent post-stroke aphasia, Fridriksson et al. [30] reported that damage to the anterior segment of the left AF seemed to negatively influence speech fluency, which implies a power predictive role of this tract in impaired speech fluency in patients with non-fluent aphasia. Coincidentally, Basilakos et al. [31] also found a relationship between damage to the anterior segment of the AF and speech fluency. In a study on patients with aphasia, Ivanova et al. [32] pointed out that the anterior segment of the left AF was related solely to production deficits. In our previous study [33], we observed a tendency for patients with anterior segment injury to the left AF to show worse recovery in terms of speech production. Similarly, the results of the current case also seem to imply a strong relationship between the damage to the anterior segment of the left AF and the poor spontaneous recovery from expressive language deficits in the subacute stage of stroke.

Moreover, several studies also highlighted the important role of the residual structure of the left AF in the recovery of post-stroke aphasia. For example, in 2013, Kim et al. [34] found that patients whose left AF could be reconstructed had a better outcome of aphasia than patients whose left AF could not be reconstructed. Similarly, Tak et al. [35] also reported that patients with discontinuation of integrity or nonconstruction of the left AF had poorer language assessment scores. Primaßin et al. [36] pointed out that preservation of the AF was an important predictor of good language recovery. In the previous study [33], we also found the importance of the residual structure of the left AF for aphasia recovery. As well, in a recent work [37], we found that the posterior segment of the left AF might be particularly crucial for supporting the residual language abilities of patients with acute/subacute post-stroke aphasia, which was consistent with the findings of Ivanova et al. [16]. Collectively, these previous findings may account for the fact that our patient underwent a good recovery of auditory comprehension, which further supports the view that an intact posterior segment is crucial for supporting residual language comprehension ability.

Generally, the evolutionary process of post-stroke aphasia is a consequence of both spontaneous recovery and the effects of speech-language therapy (SLT). Spontaneous recovery occurs mainly during the first two to three months after the onset of stroke and is deemed insufficient [38]. It was speculated that this early recovery may be a result of the restoration of blood flow and other mechanisms of tissue recovery [39]. Several reports have described a good recovery from aphemia [1,2,40], but others reported that it took a long time to recover from aphemia. For instance, Cohen et al. [5] described aphemia as the sole manifestation in a patient presenting with primary progressive aphasia (PPA). Recently, Jiang et al. [17] reported a patient with aphemia and pure word dumbness secondary to head trauma of the frontal lobe. Though traditional treatment was received for 24 days, no significant improvement was found in speech function, and the patient was unable to speak. Herein, we reported the poor spontaneous recovery of a patient with aphemia, while the follow-up MRI demonstrated an old ischemic lesion affecting the opercular part and the triangular part of the left inferior frontal gyrus. Although we did not assess changes in the function of the cortical language areas, data suggested that the recovery from aphemia was insufficient, only dependent on certain changes, including the restoration of blood flow and tissue recovery in the cortical areas.

Interestingly, in a recent study on dissociating the effect of damage to Broca’s area and neighboring brain regions on long-term speech production outcomes in 134 stroke survivors, Gajardo-Vidal et al. [41] found that damage to Broca’s area did not contribute to the long-term (at least 3 months) speech production outcome after a left frontal lobe stroke and that persistent speech production impairments after damage to the anterior segment of the AF could not be explained by a disconnection of Broca’s area. In our case, the lesion destroyed both Broca’s area and the anterior segment of the left AF, which resulted in a disconnection of Broca’s area. This supported the contribution of injury to the left AF to the poor spontaneous recovery of his speech ability. Thus, it is necessary to further observe the short-term recovery and the long-term outcome of our patient’s speech ability after consecutive speech-language therapy in the condition of the left frontal lobe being damaged. In addition, it should be noted that the lesion also damaged the anterior portion of the long segment of the left AF in our case. As it has been speculated [21] that the long segment was responsible for phonological processing and that the anterior segment might support vocalization, injury to the anterior segment might cause his persistent disability of producing voice, which masked the speech consequence of injury to the long segment, such as phonological impairment.

## 5. Conclusions

In conclusion, we report on a left cerebral ischemic stroke patient who presented with poor spontaneous recovery of aphemia that may be related to damage to the anterior segment of the left AF in addition to the consequence of the left frontal lobe lesion. Our results suggest that aphemia could be caused by the disruption of the cortical and subcortical language circuit, that damage to the anterior segment of the left AF may result in poor spontaneous recovery from expressive language deficits in the subacute stage of stroke, and that an intact posterior segment seems to be crucial for supporting residual language comprehension ability. Although the precise mechanisms underlying the poor spontaneous recovery of speech production ability in our patient need to be further clarified, this case provides new insights into our understanding of the mechanisms of aphemia. It also provides a new perspective for exploring the role of the AF in language processing. Therefore, future studies enlarging sample sizes and employing advanced neuroimaging techniques such as resting-state functional MRI or task-state functional MRI are warranted, and exploring the different extents of AF deficits in the other types of aphasia and aphemia is a significant study topic. In addition, the roles of the right AF and the other bilateral language-related white matter pathways, including the uncinate fasciculus, the superior longitudinal fascicule, the middle longitudinal fasciculus, the inferior longitudinal fasciculus, and the inferior frontal-occipital fasciculus, should also be illustrated in patients with aphemia in the future.

## Figures and Tables

**Figure 1 brainsci-12-01253-f001:**
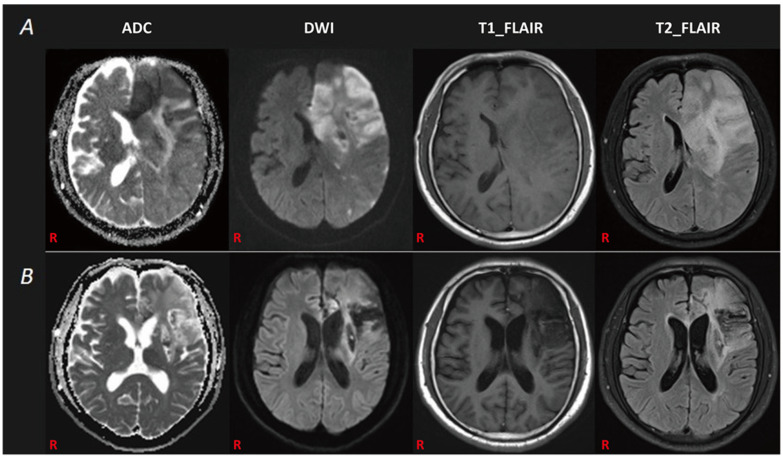
The patient’s MRI data. (**A**): The first brain MRI revealed an infarction lesion in the left cerebrum with a left basal ganglia hemorrhage. (**B**): The follow-up brain MRI demonstrated an old infarction lesion in the left cerebral hemisphere.

**Figure 2 brainsci-12-01253-f002:**
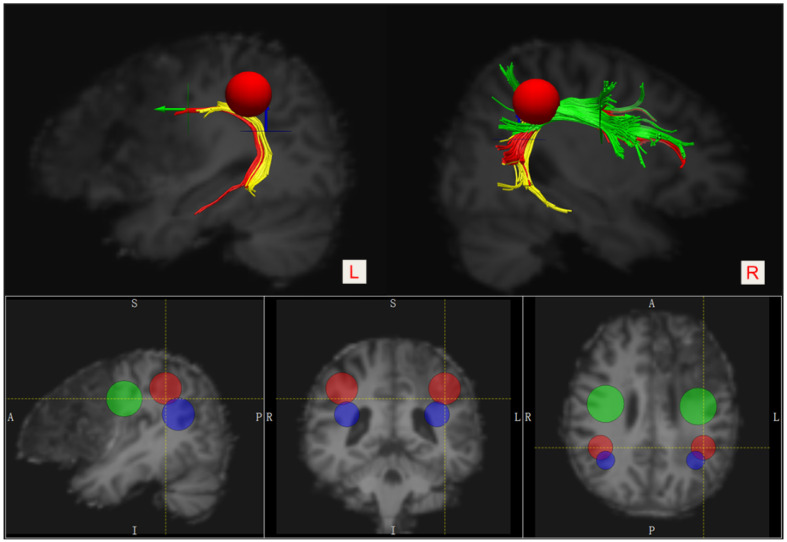
Placement of the three ROIs for visualizing segmentations of the bilateral AF. Frontal ROI—the green 2D disk, Temporal ROI—the blue 2D disk, and Parietal ROI—the red 3D sphere. The anterior segment—green, the posterior segment—yellow, and the long segment—red.

**Figure 3 brainsci-12-01253-f003:**
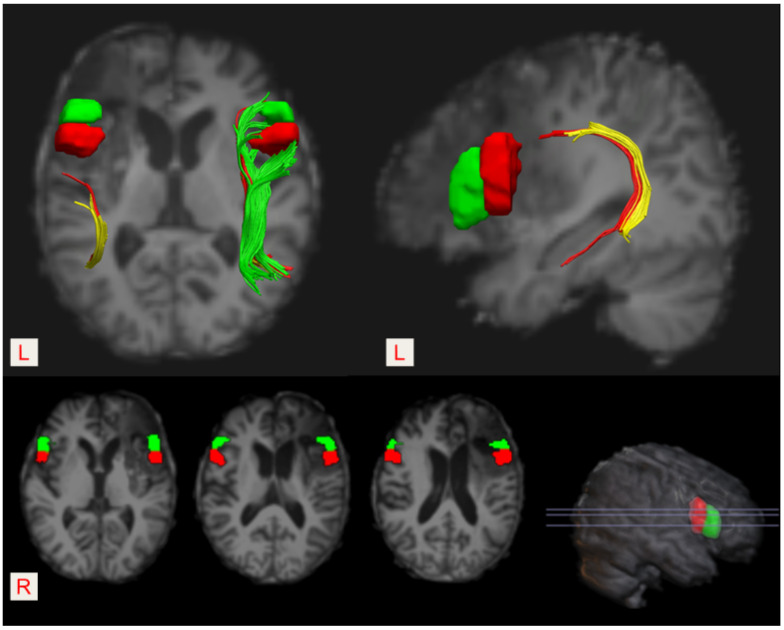
The location of the Broca’s area (pars opercularis and/or pars triangularis of the left inferior frontal gyrus) and its right homologous region. Pars opercularis—the red block, and pars triangularis—the green block. The left infarct lesion destroyed both pars opercularis and pars triangularis of the left inferior frontal gyrus.

**Figure 4 brainsci-12-01253-f004:**
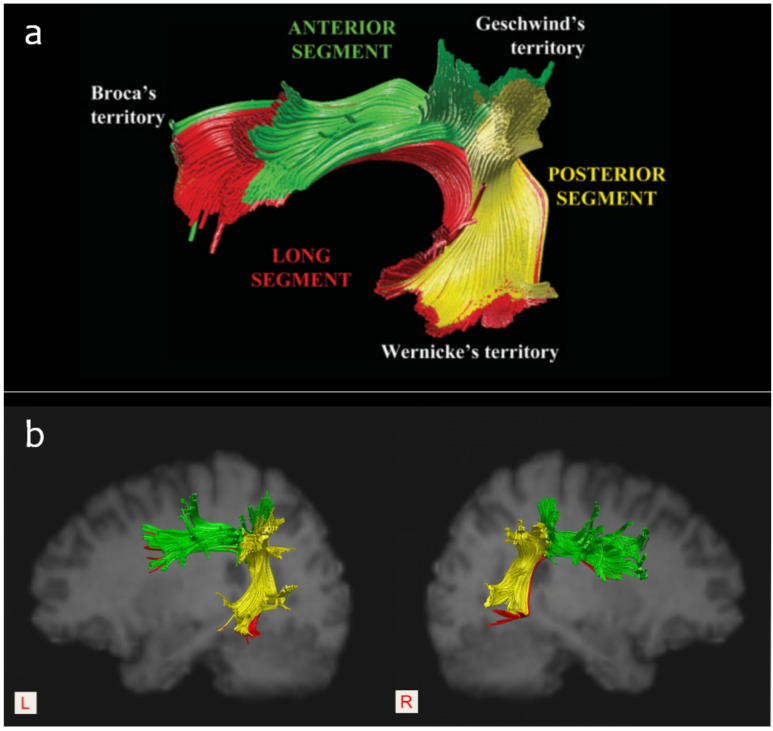
(**a**): The three-segment model of the AF outlined by Catani et al. (2005) [21], permission obtained. (**b**): A sample of the intact bilateral AF segments reconstructed according to the three-ROI approach.

## Data Availability

All data used in this study are available from the corresponding author on reasonable request.

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
