# Peer review of "Poor Spontaneous Recovery of Aphemia Accompanied by Damage to the Anterior Segment of the Left Arcuate Fasciculus: A Case Report"

_brainsci, 2022, doi:10.3390/brainsci12091253_

Round 1

Reviewer 1 Report

Dear Authors, 

I read your work entitled "Poor Spontaneous Recovery of Aphemia Accompanied by Damage to the Anterior Segment of the Left Arcuate Fasciculus: A Case Report." and here i enclose my comments:

1. The introduction is very well structured as well as your figures of your work and I congratulate them on that. 

2.  The Methods section has some weaknesses that the I suggest the Authors to address them: (a) The Authors used the MMT test, the WAB battery and the NISS screener. Are these validated in the language of the patient mother tongue, if so them provide the analogous citation, (b) Which version of WAB was administrated? The WAB has its old version and a revision called WAB-R. If its WAB-R (which includes a full sub-scale for assessing apraxia) was this battery validated in patients mother tongue? (c) Which criteria were used in order to categorise the patients as aphemic and how it was differentiated from aphasia for example? I suggest the Authors to provide that information as well. 

3. The discussion section has a pluralism but it should be re-written after the methods issues are addressed.

4. Finally, I suggest the Authors to have their manuscript checked for minor english and typing mistakes. 

Thank you.

Reviewer 2 Report

1)  The authors should state whether a one tensor or two -tensor model was used to reconstruct the DTI data

2) If possible a two-tensor DTI model should be used because of the problem with crossing fibers.

3) The damaged left side language fiber tracts should be compared to normal individual database with no brain damage so that the full extent of the AF tract can be seen.

4) Quantification of the DTI parameters such as FA, MD, RD, AD should be presented within language-related ROIs and should be compared to a non-damaged brain database (age-matched).

Round 2

Reviewer 1 Report

Dear Authors,

I read your work again and there significant updates and almost all suggestions were addressed. Its is very good that you updated the literature in the "Methods" section but I suggest you to provide the validation studies in Chinese of the assessments used.  

Thank you.

Reviewer 2 Report

The authors should add a paragraph either in the results or in the discussion citing references that show what a NORMAL/healthy tract should look like for   the anterior segment of the left AF that could not be reconstructed due to severe injury (i.e., a disconnection between Broca’s and Geschwind's area).  In other words, the authors need to better describe what a normal left AF pathway should look like based on previous literature.
